# Pushing the Envelope: Developments in Neural Entrainment to Speech and the Biological Underpinnings of Prosody Perception

**DOI:** 10.3390/brainsci9030070

**Published:** 2019-03-22

**Authors:** Brett R. Myers, Miriam D. Lense, Reyna L. Gordon

**Affiliations:** 1Department of Otolaryngology, Vanderbilt University Medical Center, 1215 21st Ave S, Nashville, TN 37232, USA; 2Department of Psychology and Human Development, Vanderbilt University, 230 Appleton Place, Nashville, TN 37203, USA; 3Vanderbilt Kennedy Center, 110 Magnolia Circle, Nashville, TN 37203, USA; miriam.lense@vumc.org; 4Vanderbilt Brain Institute, Vanderbilt University, 2215 Garland Ave, Nashville, TN 37232, USA; 5The Curb Center for Art, Enterprise, and Public Policy, Vanderbilt University, 1801 Edgehill Avenue, Nashville, TN 37212, USA; 6Department of Psychology, Vanderbilt University, 2301 Vanderbilt Place, Nashville, TN 37240, USA

**Keywords:** prosody, speech envelope, neural entrainment, rhythm, EEG

## Abstract

Prosodic cues in speech are indispensable for comprehending a speaker’s message, recognizing emphasis and emotion, parsing segmental units, and disambiguating syntactic structures. While it is commonly accepted that prosody provides a fundamental service to higher-level features of speech, the neural underpinnings of prosody processing are not clearly defined in the cognitive neuroscience literature. Many recent electrophysiological studies have examined speech comprehension by measuring neural entrainment to the speech amplitude envelope, using a variety of methods including phase-locking algorithms and stimulus reconstruction. Here we review recent evidence for neural tracking of the speech envelope and demonstrate the importance of prosodic contributions to the neural tracking of speech. Prosodic cues may offer a foundation for supporting neural synchronization to the speech envelope, which scaffolds linguistic processing. We argue that prosody has an inherent role in speech perception, and future research should fill the gap in our knowledge of how prosody contributes to speech envelope entrainment.

“In a house constructed of speech, the bricks are phonemes, and the mortar is prosody. Without the latter, we’d simply live under a pile of rocks”.—B.R.M.

## 1. Prosody Perception

Prosody is the stress, intonation, and rhythm of speech, which provides suprasegmental linguistic features across phonemes, syllables, and phrases [1,2,3]. Prosodic cues contribute affect and intent to an utterance [4] as well as emphasis, sarcasm, and more nuanced emotional states [5,6]. Certain prosodic cues are universal and can be interpreted cross-culturally even in an unfamiliar language [7,8]. Prosody also provides valuable markers for parsing a continuous speech stream into meaningful segments such as intonational phrase boundaries [9], dynamic pitch changes [10], and metrical information [11]. Parsing speech units based on prosodic perception is an imperative early stage in language acquisition, and it is considered a precursor to vocabulary and grammar development [12,13,14]. In addition, prosody can convey semantic information for context in a message [15,16]. Deficits in prosody perception have a negative downstream impact on linguistic abilities, literacy, and social interactions, e.g., [17,18,19,20].

Prosodic fluctuations are responsible for communicating a wealth of information, primarily through acoustic correlates such as duration, amplitude, and fundamental frequency. As any of these parameters changes, it influences the expression of stress, intonation, and rhythm of the spoken message [21,22]. One illustration of the dynamic and multidimensional nature of prosody is “motherese” or infant-directed speech, which is characterized by exaggerations in duration and fundamental frequency [23]. The exaggerated speech signal creates louder, longer, and higher pitch stressed syllables [24], which facilitates segmenting the speech into syllable components and disentangling word boundaries [25,26]. The modified prosodic qualities of infant-directed speech make the signal acoustically salient and engaging for infants [23,27], which yield later linguistic benefits such as boosts in vocabulary acquisition [28] and accessing syntactic structures [29]. This is one example of how prosody plays an important role in speech communication.

The importance of prosody to speech perception is widely acknowledged, yet it has been underrated in many studies examining neural entrainment to the speech envelope. The purpose for the current review is to demonstrate that prosodic processing is engrained in investigations of neural entrainment to speech and to encourage researchers to explicitly consider the effects of prosody in future investigations. We will review the speech envelope and its relation to the prosodic features of duration, amplitude, and fundamental frequency, and we will discuss electrophysiological methods for measuring speech envelope entrainment in neural oscillations. We will then highlight some previous research using these methods in typical and atypical populations with an emphasis on how the findings may be connected to prosody. Finally, we propose directions for future research in this field. It is our hope to draw attention to the role of prosody processing in neural entrainment to speech and to encourage researchers to examine the neural underpinnings of prosodic processing.

## 2. Amplitude Modulation

Prosody is determined by a series of acoustic correlates—duration, amplitude, and fundamental frequency—which can be represented in a number of ways, including the amplitude modulation (AM) envelope (also known as the temporal envelope) [30,31]. It is important to mention that a temporal waveform is composed of a “fine structure” and an “envelope”. Fine structure consists of fast-moving spectral content (e.g., frequency characteristics of phonemes), while the envelope captures the broad contour of pressure variations in the signal (e.g., amplitude over time) [32]. In other words, the envelope is superimposed over the more rapidly oscillating fine structure. Both envelope and spectral components are important for speech comprehension—i.e., to “recognize speech” rather than “wreck a nice beach” (Figure 1) (see [33] but also [34]).

It has been suggested that the extraction of AM information is a fundamental procedure within the neural architecture of the auditory system [35]. The auditory cortex is particularly adept at rapidly processing spectro-temporal changes in the temporal fine structure [36], and it is possible that this processing is aided by the amplitude envelope first laying the foundation for more narrow linguistic structure [37]. For example, fine structure cues play an important role in speech processing, yet normal-hearing listeners are able to detect these cues from envelope information alone [38]. Even when spectral qualities are severely degraded, speech processing can be achieved with primarily envelope information [39], as the envelope provides helpful cues for parsing meaningful segments in speech [40,41]. Additionally, the temporal characteristics of an auditory object allow us to focus attention on the source and segregate it from competing sources [42], which makes detection of envelope cues essential in speech communication. Because the amplitude envelope captures suprasegmental features across the speech signal, it lends itself to being an excellent proxy for prosodic information, and we argue that studies that use the speech envelope are inherently targeting a response to prosody.

The speech amplitude envelope provides a linear representation of AM fluctuations over time. Acoustic stimuli are constructed of multiple temporal dimensions [31], and modulation energy varies based upon the selected band of carrier frequencies in the signal [35]. Speech can be portrayed through a hierarchical series of AM frequency scales [43]; that is, stress placement occurs at a rate of ~2 Hz [44], syllable rate occurs around 3–5 Hz [24], and phonemic structure has a faster rate of 8–50 Hz [31] (see Figure 2). Liss et al. [45] found that energy in the frequency bands below 4 Hz was intercorrelated, and energy above 4 Hz (up to 10 Hz) was separately intercorrelated. The frequency range between 4 and 16 Hz primarily affects speech intelligibility [46], while frequencies below 4 Hz strictly reflect prosodic variations, such as stress and syllable rate [47]. These multiple timescales of modulation energy within the speech envelope have been shown to elicit corresponding modulations in cortical activity during speech processing [48]. This correspondence appears to play a role in potentially challenging listening situations, such as: speech in noise [49], multiple speakers [50], complex auditory scenes [51], conflicting visual information [52], and divided attention [53,54]. Each of these situations (discussed in more detail later) requires the listener to exploit the natural timing of speech using prosodic cues, which are provided in the amplitude envelope [55].

## 3. Neural Entrainment to the Speech Envelope

Neural entrainment to the speech envelope has been a notoriously complex topic of study for several decades. In this section we will provide a broad overview of some investigative strides in this area. It is well known that neural oscillatory activity occurs in a constant stream of peaks and troughs while at rest and during cognitive processes. This stream becomes an adaptive spike train in response to environmental stimuli, such as the acoustic signal of speech. Numerous studies have shown that neural oscillatory activity in specific frequency bands is related to specific linguistic functions; for example, lower-level linguistic processing, such as detection of stress and syllable segmentation, occurs in lower frequencies (<4 Hz) [47], and semantic/syntactic processing may occur in higher frequencies (13–50 Hz) [56].

Traditional EEG approaches to prosody perception include analyzing event-related potential (ERP) activity at key events in the speech signal [57], such as stressed syllables [58], metric structure violations [59], pitch violations [60], and duration violations [11]. While these techniques are important for determining brain responses to prosodic features, they do not provide a comprehensive measure of how the brain tracks and encodes the multidimensional aspects of prosody over time. Because prosody refers to suprasegmental features (duration, amplitude, fundamental frequency), which vary throughout an utterance, it is useful to analyze prosody across the temporal domain rather than at one point in time. For this we turn to the speech amplitude envelope as a representation of suprasegmental information.

Recent developments in the literature have explored ways to measure continuous neural entrainment, which is a phenomenon where neuronal activity synchronizes to the periodic qualities of the incoming stimuli [61]. The oscillations of the auditory cortex reset their phase to the rhythm of the speech signal, which is an essential process for speech comprehension [33]. This is known as phase-locking, which can be measured with a cross-correlation procedure between the speech stimulus and the resultant M/EEG signal [62]. Cross-correlation uncovers similarities between two time series using a range of lag windows [63]. This is an efficient method for observing the response to continuous speech without requiring a large number of stimulus repetitions, since this analysis inherently increases the signal-to-noise ratio [64].

Speech processing occurs through a large network of cortical sources [65], and phase-locking can be measured to locate functionally independent sources [66]. These sources may occur bilaterally depending on the timescale [67], such that the left hemisphere favors rapid temporal features of speech, and the right hemisphere tracks slower features. The right hemisphere generally shows stronger tracking of the speech envelope [68,69]; however, envelope tracking has also been shown to be a bilateral process [62,70].

When measuring how the speech envelope is represented in neural data, one issue with a simple cross-correlation between envelope and neural response is that temporal smearing (from averaging across time points) will create noise in the correlation function [71]. A solution to this is to use a modeling approach, known as a temporal response function (TRF) [68], to describe the linear mapping between stimulus and response. This approach stems from a system identification technique [72] that models the human brain as a linear time-invariant system. Of course, the brain does not operate on a linear or time-invariant schedule, but these assumptions are commonly accepted in neurophysiology research for characterizing the system by its impulse response [73,74].

The modeling approach can operate in either the forward or backward direction. Forward modeling describes the mapping of a speech stimulus to a neural network [68,75,76] using a TRF that represents the linear transformation that generated the observed neural signal [77] (Figure 3). When using the envelope representation of speech, the forward model treats the stimulus as a univariate input affecting each recording channel separately. However, since the speech signal is transformed in the auditory pathway into multiple frequency bands [78], the forward modeling procedure may benefit from a multivariate temporal response function (mTRF) [71], which uses the spectrogram representation to evaluate speech encoding. Even in the multivariate domain, forward modeling still maps the stimulus to each response channel independently [79].

Backward modeling is a mathematical representation of the linear mapping from the multivariate neural response back to the stimulus [71]. This modeling approach yields a decoder that attempts to reconstruct a univariate stimulus feature, such as the speech envelope. As described in [71], this decoder function is derived by minimizing the mean squared error between the stimulus and reconstruction. In the backward direction, recording channels are weighted based on the information that they provide for the reconstruction [77], which removes inter-channel redundancies—an advantage over forward modeling. By modeling in the backward direction, researchers are able to compare stimulus reconstructions to the original stimulus, for instance with a correlation coefficient as a marker of reconstruction accuracy [80]. This provides a reliable index for the degree to which the envelope is encoded in the neural network. While other methods—such as cross-correlations and inter-trial phase coherence—are adequate for measuring phase-locking in speech comprehension, the modeling approach has been gaining attention as an attractive analysis method in recent years. Regardless of the method used, measuring neural entrainment to the speech envelope is an excellent way to target prosodic processing, yet this has been underutilized in the literature.

## 4. Selected Findings in Envelope Entrainment

Many questions about speech processing can be investigated by looking at neural entrainment to the speech envelope, though we must be careful about how we interpret the results (see [34]; Table 1 provides a summary of selected studies). Peelle et al. [32] compared intelligible speech with unintelligible noise-vocoded speech, and they found that cortical oscillations in the theta (4–7 Hz) band are more closely phase-locked to intelligible speech. This may suggest that linguistic information and contextual associations enhance phase-locking to the envelope. However, others have measured envelope tracking in the auditory cortex even when the signal is devoid of communicative value. For example, Nourski et al. [81] found envelope entrainment even when speech rate was compressed to an unintelligible degree; Howard and Poeppel [82] found envelope entrainment to time-reversed speech stimuli; Mai et al. [56] found envelope entrainment to pseudo-word utterances. We acknowledge that envelope entrainment is often enhanced by intelligibility [49], but given the conflicting results described here, it is difficult to say whether intelligibility predicts entrainment or vice versa. What we can take away from these findings is that acoustic features of the stimulus—such as suprasegmental cues—seem to contribute to the neural entrainment effect and that the effect of neural entrainment on speech intelligibility warrants further investigation.

Attention has also been shown to influence envelope entrainment, and selective attention in a multi-speaker environment can be observed by the degree to which neural oscillations entrain to a given speech envelope [54,61,83]. The classic cocktail party situation has been studied for decades [84] and continues to be of interest today, e.g., [85]. In a natural auditory environment, many sounds are merged together and presented to the ear simultaneously, and the listener is tasked with segregating the sounds and attending to a particular source while ignoring the others [86]. By analyzing speech envelope representations, we can determine how the neural circuit parses and segregates these auditory objects. Ding and Simon [54] demonstrated that when a listener hears two speakers simultaneously, the neural decoding process is able to reconstruct the stimulus envelopes of both speech streams. The stimulus reconstruction is more strongly correlated to the envelope of the attended speaker (also [61,62,83]). Similar results have also been shown with invasive electrodes in electrocorticography (ECoG) research [53,87]. Despite the methodology used, these studies have suggested that neural encoding of an auditory scene involves selective phase-locking to specific auditory objects that are presented concurrently in a single auditory mixture. As mentioned previously, prosodic features of a speech stream help a listener to parse speech and attend to it, so prosody likely plays an important role in multi-speaker envelope entrainment, yet manipulations of the prosodic features of speech are rarely included as a variable in multi-speaker entrainment studies.

Speech envelopes are also of interest in studies examining audiovisual presentation of speech. Visual speech provides critical information regarding the timing and content of the acoustic signal [88]. It has long been acknowledged that listeners perceive speech better when they can both see and hear the individual speaking [89]. Articulatory and facial movements provide visual temporal cues that complement meaningful markers in the auditory stream. Visual rhythmic movements help parse syllabic boundaries [90], a wider mouth opening indicates louder amplitude [91], and seeing a conversational partner assists in segregating a speech stream from overlapping speakers [92]. Visual cues and gestures are tightly linked to speech prosody [93,94,95], and this alignment emphasizes suprasegmental features of the speech signal.

When auditory and visual information are incongruous, speech perception may be hindered and even lead the listener to falsely perceive a sound that was not presented in either modality (à la “The McGurk Effect”) [96]. Congruent audiovisual speech enhances envelope tracking compared to incongruent information and also shows greater envelope encoding than auditory only speech, visual only speech, or the combination of the two unisensory modalities [97]. Audiovisual speech also has marked benefits for neural tracking when presented in noisy conditions [98] (also [99,100]). This is indicative of multisensory enhancement during speech envelope encoding. At the same time, there appears to be a similar mechanism for visual entrainment in which cortical oscillations entrain to salient lip movements even when they are incongruous to the acoustic stream [101]. These studies of envelope responses to speech incongruence support an emerging model of correlated auditory and visual signals dynamically interacting in a discrete process of multisensory integration [88,102]. Prosody is a major factor in this integration, as it aligns a stable framework of temporal and acoustic–phonetic cues to be used in speech processing; however, the contribution of prosodic dimensions of the speech stimuli to neural entrainment in multisensory processing in these studies has not been explicitly considered.

Prosody shares a number of features with music, so an area for potential exploration is the connection between neural entrainment to speech and to music. Envelope entrainment is influenced by speech rhythm [103]. Because rhythm and temporal cues provide a common link between music and speech perception (e.g., [104,105]), several studies demonstrate associations between musical rhythm aptitude, speech perception, and literacy skills in children [106,107]. Some have hypothesized that entrainment to music leads to increased timing precision in the auditory system, which leads to increased perception of the timing of speech sounds [108,109]. Doelling and Poeppel [110] found that the accuracy of cortical entrainment to musical stimuli is contingent upon musical expertise, suggesting individual differences in cortical oscillations related to experience. However, musical expertise does not necessarily predict stronger entrainment to the speech envelope [111]. Additional work on individual differences between speech and music may help to target the neural mechanisms behind prosodic processing.

In summary, neural entrainment to the speech envelope likely reflects, at least in part, prosody perception. Prosodic fluctuations and prosody perception likely contribute to experimental findings linking envelope entrainment to intelligibility, selective attention, and audiovisual integration. Findings discussed in this section are highlighted in Table 1.

## 5. Developmental and Clinical Relevance of Envelope Entrainment

Children show a reliance on prosody processing from early infancy [122,123]; so, the envelope appears to be a critical tool for early language acquisition. The speech amplitude envelope contributes to the perception of linguistic stress, providing essential information for speech intelligibility and comprehension, e.g., [124]. Infant-directed speech is a manner of speaking that exaggerates prosodic cues, and infants show stronger cortical tracking of the infant-directed speech envelope compared to tracking of adult-directed speech [125]. Individuals who have difficulties with processing cues related to the speech amplitude envelope may demonstrate language-processing deficits [126].

Neuronal oscillatory activity in healthy adults entrains to adult-directed speech at various timescales, e.g., [33]. Frequencies in the delta band range (1–4 Hz) involve slower oscillations and track suprasegmental features of speech, such as phrase patterns, intonation, and stress [33,61]. Prosodic cues are particularly salient in the delta band and may be of particular relevance for envelope entrainment and language acquisition in children. Child-directed speech appears to bolster entrainment at the delta band specifically by amplifying these prosodic features [127]. The accuracy of delta band entrainment may also be indicative of higher-level linguistic abilities, as entrainment at the 0–2 Hz band is positively correlated with literacy [115,128]. The delta band may be crucially important because it provides the foundation for hierarchical linguistic structures of the incoming speech signal [33]. This could, in turn, affect cross-frequency neural synchronization, which may be particularly informative for the development of speech comprehension [129]. 

Autism spectrum disorders (ASD) are associated with atypical processing of various sensory modalities [130]. Individuals with ASD show less efficient neural integration of audio and visual information in non-speech [131] and speech input [132]. This is related to the temporal binding hypothesis in ASD, which suggests that these individuals have a deficit in synchronization across neural networks [133]. Jochaut et al. [134] showed deficient speech envelope tracking using fMRI and EEG when individuals with ASD perceive congruent audiovisual information. Possible impairment in coupling rhythms into oscillatory hierarchies could contribute to these results [135], and examining language deficits in ASD as oscillopathic traits may be a promising step forward in understanding these disorders [136,137].

Developmental dyslexia is a disorder of reading and spelling difficulties not associated with cognitive deficits or overt neurological conditions, and it is often considered a disorder of phonological processing skills [138]. Dyslexia is believed to affect the temporal coding in the auditory and visual modalities [139,140], and individuals with dyslexia often have difficulty identifying syllable structure or rhyme schemes, see [141]. The speech envelope is important to study in dyslexia because it carries syllable pattern information, and Abrams et al. [142] reported delayed phase-locking to the envelope in individuals with dyslexia. Specifically, the delta band in neuronal oscillations can reveal anomalies such as atypical phase of entrainment [43,143] and poor envelope reconstructions [115], which may ultimately have a downstream effect on establishing phonological representations [25]. Because the delta band reflects prosodic fluctuations, the atypical entrainment in this range suggests that individuals with dyslexia may have impaired encoding at the prosodic linguistic level [61].

Developmental language disorder (DLD) affects language abilities while leaving other cognitive skills intact, and it is sometimes studied in parallel with dyslexia due to similar deficits in phonological and auditory processing [144,145]. The prosodic phrasing hypothesis [146] suggests that children with DLD have difficulty detecting rhythmic patterns in speech, particularly related to impaired sensitivity to amplitude rise time [147] and sound duration [126], and difficulties in processing accelerated speech rate [148]. Given the growing behavioral evidence suggesting that children with DLD have deficits in prosody perception (see [149]), it stands to reason that they would show poor speech envelope entrainment, particularly in the delta frequency band [33]. To our knowledge, there has not been an electrophysiological study looking at neural entrainment to the speech envelope in children with DLD, but this would be an illuminating endeavor.

## 6. Directions for Future Research

There have been many recent advances related to speech envelope entrainment, and we argue that prosody has had a substantial—though at times underrated—role in many studies. It is well accepted that prosodic cues facilitate speech processing, e.g., [3], and these suprasegmental features are represented in the amplitude envelope, e.g., [64]. Therefore, studies investigating speech envelope entrainment inherently capture a response to prosody to some degree, yet the underlying mechanisms of prosody perception, and their effect on speech processing, remain somewhat a mystery. We suggest that including experimental manipulations of the prosodic dimensions of speech in future studies may inform the findings of previous works, and it may shed light on the future interpretation of entrainment, particularly in the low-frequency range. Ding et al. [118] have shown that removing prosodic cues from speech weakens envelope entrainment, which suggests that synchrony between neural oscillations and the speech envelope reflects perception of the acoustic manifestations of prosody, and future work should continue testing this relationship. More broadly, we present a series of potential future directions in Table 2.

Synchronization occurs when internal oscillators adjust their phase and period to rate changes of speech rhythm, e.g., [150]. According to the dynamic attending theory [151,152], attentional effort is not uniformly distributed over time, but rather, it occurs periodically with salient sensory input. Prosody offers meaningful information through stressed syllables, which gives attentional rhythms a structure for scaffolding speech processing mechanisms [108]. Suprasegmental elements are present in a wide array of stimuli that demonstrate neural entrainment to the speech envelope (e.g., intelligible and unintelligible speech; attended and unattended speech; audiovisual and audio only and visual only speech). The suprasegmental cues may be one reason why stimuli of varying salience continue to reveal entrainment. Future empirical investigations may consider how prosody supports neural entrainment under these different experimental conditions. 

Of course, prosodic fluctuations alone cannot fully explain neural entrainment to speech or speech comprehension [34]. When Ding and colleagues [118,153] removed prosodic cues from connected speech stimuli, they did find some low-frequency entrainment (<10 Hz), which they attribute to syntactic processing. However, they pointed out that neural tracking would likely be more prominent in natural speech with the addition of rich prosodic information. In spoken English, syntax can exist without prosody, but the inclusion of prosody certainly facilitates syntactic processing (with phrase segmentation, pitch inflection, etc.). Therefore, further study of prosodic versus syntactical manipulations will shed light on their respective contributions—and their interaction—to neural entrainment to speech, including when examined together with behavioral measures of speech comprehension. Studies have shown that phonetic [117] and semantic [120] levels of processing also contribute to neural activity at different hierarchical timescales. It may be informative to consider how prosodic cues organize and facilitate processing at these different levels. Future work should attempt to isolate prosodic cues from phonetic and semantic details to specify the contributions of prosody to these other structures in continuous speech. This could be accomplished by restricting prosodic cues (using monotone pitch and constant word durations, as in [118]) or by creating stimuli with a prosodic mismatch (using unpredictable changes in amplitude, pitch, and duration). These manipulations would allow researchers to more directly target the role of prosody in entrainment.

Examining the links between prosody and neural encoding of the speech envelope may also have relevance for additional topics and clinical populations. For example, it has been shown that features of prosody are directly linked to emotional expressiveness in speech [6], and one novel area of research would be to connect patterns of envelope entrainment with perception of emotional states. This would likely have implications for the clinical populations discussed above, as well as typical emotional development. Other recent work has investigated rhythmic cueing and temporal dynamics of speech in patients with Parkinson’s disease [154], aphasia [155], and even blindness [156]. Because these populations show difficulty with prosodic cues in speech, a next step could be to examine speech envelope entrainment in these individuals to examine if there is a neural deficit in prosody encoding.

As conveyed in this review, low-frequency neural oscillations likely reflect in part a response to prosodic cues in speech. Future research can investigate how prosody impacts neural envelope entrainment and scaffolds higher-level speech processing, as well as examine individual differences in prosody perception and neural entrainment. Future research in speech entrainment ought to search for connections to prosody perception and determine what it takes to get the speech envelope signed, sealed, and delivered to the cortex.

## Figures and Tables

**Figure 1 brainsci-09-00070-f001:**
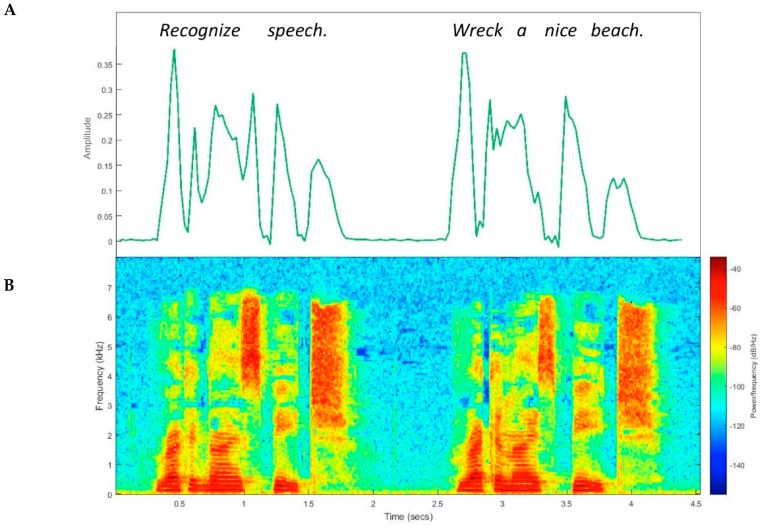
Two representations of an acoustic speech signal: Amplitude envelope (**A**) and spectrogram (**B**). Subtle differences between the phrases “recognize speech” and “wreck a nice beach” can be detected in both representations.

**Figure 2 brainsci-09-00070-f002:**
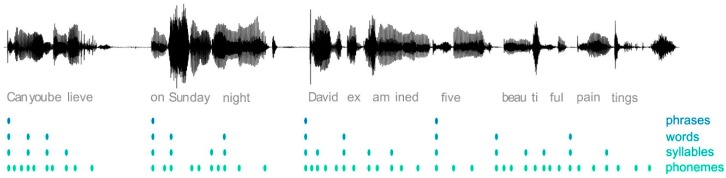
Acoustic waveform with its segmentation into phrases, words, syllables, and phonemes. Figure reproduced from [48].

**Figure 3 brainsci-09-00070-f003:**
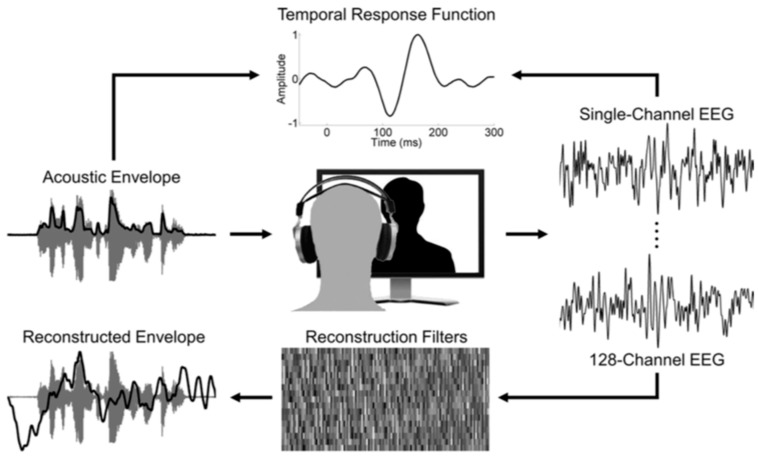
The temporal response function (TRF)—calculated with a linear least squares approach—represents the mapping from acoustic envelope onto each channel of EEG data (forward modeling). A multivariate reconstruction filter can be applied to data from all channels to estimate the acoustic envelope (backward modeling). Reconstruction accuracy can be measured by Pearson correlation between original and reconstructed envelopes. Figure reproduced from [75].

**Table 1 brainsci-09-00070-t001:** List of papers investigating speech envelope tracking using various analysis approaches, data collection procedures, and topics of interest. Analysis abbreviations: CC—cross-correlation; PC—phase coherence; TRF—temporal response function; SR—stimulus reconstruction.

Author/Year	Data	Analysis	Relevant Amplitude Envelope Findings
**Speech Intelligibility**			
Ahissar et al., 2001 [63]	MEG	CC	Phase-locking predicts speech comprehension
Luo and Poeppel, 2007 [112]	MEG	PC	Phase-locking to speech is robust at 4–8 Hz
Abrams et al., 2008 [69]	EEG	CC	Right-hemisphere dominance for phase-locking
Hertrich et al., 2012 [64]	MEG	CC	Phase-locking with right-lateralized peak at 100 ms
Ding and Simon, 2013 [49]	MEG	TRF	Phase-locking at <4 Hz remains stable in noise
Peelle et al., 2013 [32]	MEG	PC	Phase-locking is strongest at 4–7 Hz in intelligible speech
Ding et al., 2014 [113]	MEG	TRF/PC	Phase-locking at 1–4 Hz predicts speech comprehension
Millman et al., 2015 [114]	MEG	CC	Phase-locking at 4–7 Hz regardless of intelligibility
Power et al., 2016 [115]	EEG	SR	Reconstruction of vocoded speech is strongest at 0–2 Hz
**Cocktail Party**			
Power et al., 2012 [61]	EEG	TRF	Attention elicits left-lateralized peak at 209 ms
Ding and Simon, 2012 [54]	MEG	SR	Attended speech phase-locks at <10 Hz around 100 ms lag
Zion Golumbic et al., 2013 [87]	ECoG	PC	Attended speech phase-locks at 1–7 Hz and 70–150 Hz
Horton et al., 2014 [50]	EEG	CC	Attended phase-locking improves with sample length
O’Sullivan et al., 2015 [83]	EEG	SR	Attended speech encodes maximally at 170–250 ms lag
O’Sullivan et al., 2017 [116]	ECoG	SR	Attention boosts reconstruction accuracy in dynamic switching
**Audiovisual Speech**			
Crosse et al., 2015 [97]	EEG	SR	AV speech encodes better than A + V at 2–6 Hz
Crosse et al., 2016 [98]	EEG	SR	AV speech improves reconstruction in noise at <3 Hz
Park et al., 2016 [101]	MEG	PC	Cortical activity entrains to lip movements at 1–7 Hz
**Linguistic Information**			
Di Liberto et al., 2015 [117]	EEG	SR	Cortical activity entrains to phonetic information
Ding et al., 2017 [118]	EEG	PC	Cortical activity entrains to multiple levels concurrently
Falk et al., 2017 [119]	EEG	PC	Phase-locking improves when rhythmic cue precedes speech
Broderick et al., 2018 [120]	EEG	TRF	Neural tracking depends on semantic congruency
Makov et al., 2017 [121]	EEG	PC	Phase-locking at 4 Hz during sleep, but not at higher levels

**Table 2 brainsci-09-00070-t002:** Potential future directions including key points and methodological considerations.

Future Directions	
Key Point	Potential Directions and Methodological Considerations
Prosodic characteristics of stimuli should be controlled and well-described	Is it feasible to equalize the prosodic dimension across experimental conditions that are not meant to isolate prosody? At the least, authors could describe the metrical structure of speech stimuli in studies that examine entrainment to envelope features. What is the variability of neural entrainment to stimuli that differ in prosodic structure?
Role of repetition in establishing neural entrainment to prosodic cues	What are the implications of hearing the same sentence/stimulus repeated many times versus hearing novel speech? Repetition affects semantic and syntactic expectancies, as well as expectations for the unfolding envelope of the signal. What is the relationship between predictive neural processes, entrainment to the envelope, and intelligibility? How do these concepts relate to prosody?
Low-frequency envelope fluctuations correlate with the syntactic structure of speech and are relevant to language development	Does entraining to envelope phrase boundary markers (such as pauses and phrase-final lengthening that correlate with important syntactical information) explain variance in syntactic processing? How does detection of these cues evolve over the course of childhood language development? Is neural entrainment to the envelope a potential signature of development of sensitivity to these cues?
Individuals vary in their sensitivity to prosody	Does neural entrainment to the envelope reflect how individuals differ in their prosodic sensitivity (when measured as a separate behavioral trait)? Can environmental and genetic factors such as musical training and music aptitude affect individual differences in neural entrainment to speech? Do some individuals with developmental disabilities have impaired neural entrainment to the envelope? How does this differ among different neurodevelopmental disorders? Is there a causal impact of this impairment on their speech/language/reading development? Can speech/language/reading therapy enhance sensitivity to prosody via increased neural entrainment to the envelope (as a mediating mechanism to improving speech/language/reading outcomes)?

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
