# Peer review of "Pushing the Envelope: Developments in Neural Entrainment to Speech and the Biological Underpinnings of Prosody Perception"

_brainsci, 2019, doi:10.3390/brainsci9030070_

Round 1
Reviewer 1 Report
Summary
This review aims to highlight the role of prosody in speech perception and the lack of empirical neurophysiological investigations into its neural underpinnings. In particular, the review focuses on the use of envelope tracking measures as a means to probe natural speech processing in electrophysiology. The paper reviews the recent envelope tracking literature and the resulting advances in our understanding of speech processing as well as selective attention and multisensory integration, linking it to prosodic information processing. Finally, the paper discusses the potential of this approach to elucidate on the neural underpinnings of various developmental and clinical disorders and discusses the future directions that the field should take in order to dos so, bearing in mind the role of prosody.
The role of prosody perception in speech processing has indeed been much neglected in many studies that examining neural entrainment to the speech envelope. But its importance to speech perception is in no doubt, thus this review represents an important call to researchers to fill in the gaps in our knowledge regarding how prosody contributes to envelope entrainment. The review starts off very strong and is extremely well written, emphasizing the relevance of the article quite well. The first section on prosody perception sets up the review nicely, is informative and has a really nice flow. I really enjoyed reading it and learned quite a bit! However, I am regretful to say that the quality of the review declined from there on for reasons that I outline below. As such, I recommend that the manuscript is rejected in its current state but hopefully can be accepted at a later date with some major revisions. I have given as much advice as I can in the minor comments section in the hope that it will help the authors improve the manuscript.
Major comments
As indicated by the number of minor comments below, the section on Neural Entrainment to the Speech Envelope in particular, contained many inaccuracies in terms of the literature and terminology which did not match the quality of the previous sections. I would suggest some careful reading of seminal reviews on encoding/decoding models and system identification by the likes of (Wu et al., 2006; Haufe et al., 2014; Holdgraf et al., 2017).
I felt there were large sections of the article that got lost in terms of direction. As if they were written as part of something else and joined together with a token unifying statement at the end to bring the reader’s attention back around to prosody perception. I feel the main argument of the article needs to be better woven into the fabric of all the sections, and not just the first. This may take substantial rewriting but will be worth it and improve the quality and flow of the article immensely. Look no farther than the first section for inspiration.
While the main aim of the review was to call researchers to call researchers to consider the role of prosody in their investigations, I felt that the authors could have done more to provide details as to how this could be achieved, particularly in the final section. Proposing specific details about future approaches would greatly enhance the value of such work. What kind of empirical evidence is needed? Are there any theoretical or computational models that would be useful to consider?
Minor comments
There is no reference to the table in the text. Also, paper by Crosse and Lalor (2014) retracted as effect was due to systematic error.
Abstract: try to link the second half of the abstract which talks about envelope tracking back to prosodic processing.
In the speech literature, the amplitude envelope is more commonly referred to as the temporal envelope, in order to distinguish it from a spectrogram representation. The fact that it reflects fluctuations in intensity or amplitude is a given.
Line 82: What do the authors mean by a linear representation? Are they referring to the fact that the broadband envelope is the sum of the envelope at each frequency band?
Line 87-88: What do the authors mean by energy in the power spectrum below 4Hz was intercorrelated? As in the power at different narrowbands below 4Hz?
Line 111-112: This statement contradicts your earlier statement where you relate the temporal scale of linguistic units to particular neural oscillatory frequency bands. My understanding is that higher frequencies reflect processing of more fine-grained segmental units (phonemes, morphemes, syllables etc.) and lower frequencies more course-grained suprasegmental units (words, phrases, sentences etc.). This also makes sense when you think about how ERP research has characterized the timecourse of speech evoked AEPs, relating longer latencies (which correspond to lower frequencies) to higher-level linguistic processing (phonetic, lexical-semantic etc.). See Salmelin (2007).
Line 119: I’m not sure if I would use the term ‘periodic qualities’ to describe the properties of speech that neuronal populations entrain to. While the segmental and suprasegmental units of speech tend to have specific temporal properties, they do not unfold within a given sentence in a periodic manner necessarily, even if they always maintain their temporal properties. The rhythm of speech is highly variable, but at the same time predictable due to context and prior knowledge. I would use the term ‘temporal properties’ instead.
Line 119-121: What about an evoked account of envelope tracking? In other words, manifesting as a result of continually evoked, overlapping impulse responses (i.e., convolution). Importantly, one of the assumptions that system identification makes is that the response is a convolution of the input and some unknown response function.
Line 123: I wouldn’t say “using a range of lag windows” unless you are specifically talking about using a lagged time-series representation as you do in SI. Cross correlation simply uncovers similarities between two time-series at every possible lag (positive and negative).
Line 132-134: Smearing occurs because speech is correlated with itself at multiple non-zero time lags, thus it will be correlated with the EEG and multiple overlapping time lags resulting in a broadened or smeared cross-correlation estimate. The cross-correlation procedure is agnostic to the autocorrelation properties of speech and assumes the speech stimulus at a given time point must generate fluctuations in the EEG at any time lag that is temporally coherent with changes in the stimulus.
Line 134-135: The approach itself is not known as a temporal response function (TRF), rather the linear stimulus-response mapping function is known as a TRF (but only in the forward direction). The approach can be referred to as (among other things) forward modelling, regularized linear regression, normalized reverse correlation.
Line 137-139: Also, non-invasive (EEG) studies have shown that use of more complex non-linear approaches does not improve modal performance significantly (Crosse, 2011; Power et al., 2011a, b) which may be due to the diversity of non-linear responses across neuronal populations making them less detectable by non-invasive recording such as EEG (Mesgarani et al., 2009; Crosse et al., 2016b).
Line 143-144: Multivariate reconstruction filter has to be estimated first using backward modelling approach before applying to new EEG data to reconstruct the envelope.
Line 143-145: Reconstruction accuracy can be measured a number of different ways, not just Pearson’s r. Could use different measures of correlation (e.g., Spearman, Kendall, Rankit) and also error measures such as root mean squared error (RMSE). Better to say “Reconstruction accuracy can be measured by Pearson’s…”.
Line 146-148: This sentence says the same thing twice using different language. Forward modeling is the approach, and TRF is what quantitatively describes the mapping. TRF can also be referred to as a forward model. Note difference between ‘a forward model’ and ‘forward modeling’.
Line 155: Backward models do not use ‘TRFs’. It’s the same mathematics, but with the input and output reversed. Because TRF stands for temporal response function, by definition, it refers to the impulse response of the system, which can only be described in the forward direction as that is how the response is generated. Backward models (or reconstruction filters or decoders), are just mathematical representations of how the stimulus can be inferred from the responses, but in reality the responses do not cause or generate the stimulus. It’s just a useful method for quantifying stimulus encoding.
Line 1156-157: I wouldn’t use the term generate when describing backward modeling to avoid confusion with forward models which are by definition generative models. Maybe use ‘yields’ or ‘derives’.
Line 158: Correction: “this decoder is derived by minimizing the mean squared error between the stimulus and reconstruction”.
Line 160: It’s not the contribution-dependent channel weighting that “does not allow inter-channel redundancies”, it’s the fact that in the backward model, the autocovariance structure is now based on the response (RTR), meaning the inter-cannel correlations are divided out of the model, revealing only unique contributions at each channel.
Line 162-163: Again, there are a number of ways to evaluate reconstruction accuracy, not just Pearson’s r.
Line 163: I would use “reliable index” as opposed to “efficient representation”.
Line 148-169: I agree, see Ding et al. (2014) theory of an analysis-by-synthesis mechanism and its relationship to envelope tracking.
Line 178-179: True, but there are studies demonstrating strong relationships between envelope tracking and speech intelligibility (Ding and Simon, 2013). So while entrainment may be primarily driven by acoustic features, it is also enhanced by intelligibility in both a bottom-up manner (signal quality, background noise) and a top-down manner (prior knowledge, attention).
Line 186: “Neural circuit” does not convey the broad, distributed network of brain areas that are likely involved in speech processing and auditory scene analysis.
Line 196-199: These inverted peaks that suppress the unattended speaker have also been demonstrated by O'Sullivan et al. (2015).
Line 200: I wouldn’t refer to the forward modelling approach in Power et al. (2012) as AESPA, this term was used by the authors because of their previous research using white noise (spread spectrum) stimuli. It’s the exact same thing as a speech envelope TRF so can be referred to as such.
Line 201-202: O'Sullivan et al. (2015) did not replicate the study by Power et al. (2012), they actually re analyzed the data from the latter study, but using a backward modelling approach instead of forward modelling.
Line 225: I wouldn’t describe this as the “summation of the two modalities” but rather just a combined unisensory or non-integrative multisensory account.
Line 227-229: Why a distinct mechanism? Congruent lip movements also implicitly encode information about the acoustic envelope (Crosse et al., 2015b; O’Sullivan et al., 2017) that is highly predictive of intelligibility or lip-reading ability (Crosse et al., 2015a). Why would this not be a similar process to auditory envelope tracking just within a different sensory system?
Line 230-231: How can A and V signals interacting dynamically be a discrete process?
Line 246: Certainly from a cross-modal perspective, the role of prosodic information has been highlighted. Munhall et al. (2004) showed that delta frequency head movements, which have been shown to convey prosodic information, are important to speech intelligibility and Megevand et al. (2018) recently showed delta-band phase resetting of auditory cortical oscillations by visual speech. Other studies (Ding and Simon, 2013; Ding et al., 2014; Crosse et al., 2016a) have demonstrated the robustness and importance of delta-frequency activity (which likely reflects prosodic information) for entraining to speech-in-noise.
Line 276: See also Foxe et al. (2015) for compelling example of AV speech integration deficit in ASD.
Line 276-278: This may not be exclusively related to the so-called temporal binding hypothesis. It could be do with disrupted connectivity (Vasa et al., 2016), impaired cross-modal inhibition (Murphy et al., 2014) or even a delay in the developmental shift from multisensory competition to multisensory facilitation (Cuppini et al., 2018) due to lack of multisensory experience (Cuppini et al., 2017).
Line 291-293: It’s not clear to me why considering “cross-frequency neural synchronization” would “help identify the neural underpinnings of impaired phonology” in dyslexia.
Line 316: typo
Line 320-321: See new empirical evidence in support of this using psychophysics (Fiebelkorn et al., 2013) and electrophysiology (Fiebelkorn et al., 2018; Helfrich et al., 2018).
Line 336: “hierarchically-organized time scales”.
Line 341: Can the authors expand on how to consider this relationship.
Line 346: “relationship between”
Author Response
The authors review the literature on neural entrainment to speech envelope, while developing the argument that studies in this field have missed the extent to which prosodic processing may be captured in the analysis of neural entrainment.
General assessment
The general topic of neural entrainment to the speech amplitude envelope is timely and undoubtedly in need of integrative/critical reviews. In this regard, the authors did an excellent choice of topic, and they also did a massive work of literature review. I think, however, that the main argument (prosodic processing is missing in the investigation of neural entrainment to speech) is not consistently presented, partly due to lack of clarity in concept definition. I also found that the paper seems to lack direction – as if it was built from a concatenation of fragments, instead of following a clearly defined plan, both at the micro and macrolevels. Overall, I am afraid this review may confuse the reader more than inform him/her. Since I am not sure that these structural problems can be easily revised (specially within the time frames of this journal), I would suggest rejection of this paper. I hope this does not discourage the authors from moving on with this potentially valuable work.
Thank you for this feedback. As you will see below, we have significantly revised the paper to follow a clear structure that supports our main argument. We feel that the direction of the paper is improved and believe that it will be clear, interesting, and useful to the readership of this journal.
Lack of clarity in concept definition
Speech amplitude envelope – It is defined as “the broadband contour of fast-moving spectral content (“fine-structure”), and it encapsulates important temporal features of the acoustic signal” (ln 68). Ahead, they mention that it captures “features of timing, pitch and loudness” (ln 80). My opinion is that this definition is vague, since it does not clearly exclude anything in the signal. The problem starts with “fast-moving spectral content”: what does it mean – F0 and higher-than-F0- frequencies, including formants and formant transitions (spectrum in a strict sense)? Does it relate to segmental information (implying formant and formant transition detection), suprasegmental (relying on F0), both?
We agree that the speech amplitude envelope definition was vague, and we have redefined it in clearer terms (ln 70-78). Later (ln 93-96) we reiterate that the envelope relates to suprasegmental information.
Spectral vs. temporal information – Related to the previous, definitions for these terms are never provided, although they are a target for discussion (ln 76-78). Moreover, how do spectral and temporal levels relate to the speech envelope? Does the latter contain both (see ln 73-81)?
We clarify this distinction in our definitions of envelope and fine structure (ln 70-78). The speech envelope is generally thought of as a carrier of temporal information, but as discussed in this section, listeners can also detect spectral information from the envelope (ln 87).
Prosody – Although the authors provide a definition early in lns 34, assigning three dimensions to prosody (stress, intonation and rhythm), in the remainder of the text they refer to prosody as an apparently unidimensional object. This raises obstacles in the understanding of the main argument, as I will try to show next.
Prosody is a concept that is notoriously difficult to define, and it is not our intent to convey it as unidimensional. We describe it as a multifaceted entity that consists of several dimensions (ln 49)—much like how the envelope consists of complex variations in fine-structure. We hope this has become clearer in our revised manuscript.
Main argument not consistently presented
-The main argument is that prosodic processing is absent in the investigation of neural entrainment to speech. However, the authors mention that the speech amplitude envelope reflects stress, syllable rate, and phonemic structure (lns 82-87). Getting back to the authors’ definition of prosody, I would say that prosody is inherently targeted by studies of entrainment to speech envelope since stress and rhythm (syllable rate) are conveyed by the latter. This is stated explicitly in lns 90-91 (see also ln 241, where the authors write that the speech envelope reflects rhythm). The exception would be intonation. Could it be that the authors are referring to the intonation dimension of prosody, when they refer to the “missing link” (suggested perhaps in lns 100-101)? If it is so, I guess it should a priority to make it clear. If it is not the case, then the argument does not make much sense, unless I am seriously missing something (this is a possibility that I leave open to the editor’s judgement).
Yes, this is exactly the main argument—that studies of entrainment to the speech envelope are inherently targeting prosody. However, these studies seldom discuss the contribution of prosodic cues to their results, and this paper urges future studies to do so. We have revised the manuscript to reiterate this throughout, including the revised final paragraph in the first section (ln 57-68).
-In ln 58, the authors state their argument and say that the speech envelope may be linked to prosody contour. Again, what do they mean by prosody contour? Is it intonation? Because if the term relates to all prosodic aspects, we will soon be presented with findings that the speech envelope relates to stress and rhythm (lns 85-87), providing the answer to the authors’ question.
We have removed the term prosody contour, as this is not clearly defined in the speech literature. We have clarified that the envelope relates to all prosodic aspects of speech, including duration, amplitude, and fundamental frequency (ln 70).
-the idea of entrainment to different frequency bands, reflecting different linguistic levels, is a core aspect in this review. However, there are apparent inconsistencies which are never discussed (or else non-clarified). The authors start by defining a band in the speech envelope below 4 Hz to establish the realm of prosodic traces (lns 90-91), and another from 8 to 50 Hz to define phonemic content (ln 87). Later on (ln 111), they say that acoustic-phonetic processing occurs in lower frequencies (<8Hz). How does this all integrate? Are the authors referring to something other than neural entrainment to the speech envelope in lns 111?
We have rephrased these sections to consistently describe frequencies below 4 Hz as relating to prosodic fluctuations (ln 104 and 123).
Lack of direction (micro and macro)
-I guess the obvious structure for a paper like this would be (a) to show that studies of neural entrainment to speech envelope have contributed to understand a number of processes in speech comprehension but (b) they have failed to do so concerning prosodic processes (or is it intonation?...), even though it should be possible. The authors move away from this obvious structure in several ways, making it very difficult to follow the big picture. The main argument (see above) is presented several times, under different shapes and contexts (e.g., ln 58, ln 100, lns 244-251, 338-339,340-344), and it is hard to understand what motivates the question subtending the main argument. At the end of the day, it feels like asking: what have researchers done till now? What have they missed? Did they actually miss prosody? Is the problem that the authors identify out actually a problem?
We agree that the structure of the paper was not obvious at first. However, we have revised the paper significantly to follow what we hope is a clear structure that supports the main argument. The paper is organized (a) to discuss prosody and its role in the speech amplitude envelope, (b) to highlight methods in electrophysiology that have incorporated the envelope, (c) to give an overview of studies that use the envelope and demonstrate that prosody may be a key factor even if not explicitly mentioned, and (d) to demonstrate that prosody has a clear role in these studies and should be addressed more explicitly in the literature. All of this feeds into the main purpose, which is to encourage researchers to fill the gap in understanding how prosody contributes to envelope entrainment.
-instead of using the obvious structure (see above), the authors expand into topics such as the effects of attention or cross-modal integration in neural entrainment: how does this relate to the missing link of prosody? They also emphasize the importance of neural entrainment in a number of pathologies, but the extent to which these relate to prosody is unclear.
We present these findings to provide a review of recent topics in this area, and to point out that prosody is inherently important to all of these studies, yet it goes unmentioned. We have revised these sections to explicitly show how prosody fits into this work, and we hope this has become clearer.
-at the microlevel, I found a number of gaps in the authors text sequencing and also instances of poor phrasing, such as:
Ln 35: prosodic cues contribute affect? – typo?
Here we refer to “affect” (the noun) as the state of emotion conveyed by prosody.
Ln 45: is it due to the suprasegmental nature of prosody that one extracts a wealth of information? – poor phrasing
We have revised this phrasing (ln 46).
Ln 48: the topic of child-directed speech is introduced abruptly. Maybe the authors want to present it as an illustration of the role of prosody, but it does not come like that to the reader (sequencing).
We have revised this with a smoother and logical transition (ln 49).
Ln 55- these acoustic properties….Which properties? I guess they are in ln 47, but they are too far away at this point (sequencing).
This sentence has been removed in our edits, but we were referring to duration, amplitude, and fundamental frequency (ln 70).
Ln 58 -60: The authors argue something related to the speech envelope without having mentioned this term before (sequencing).
We agree that this statement seemed out of place, and we have removed it, reserving the speech envelope discussion for the second section.
Ln 73-75: I cannot understand this sentence, namely the word “but” (sequencing).
This sentence has been removed in our edits.
I hope my comments will help the authors to publish in the near future, and I wish them the best of luck.

Reviewer 2 Report
Review
Myers, Lense and Gordon (2019)
The authors review the literature on neural entrainment to speech envelope, while developing the argument that studies in this field have missed the extent to which prosodic processing may be captured in the analysis of neural entrainment.
General assessment
The general topic of neural entrainment to the speech amplitude envelope is timely and undoubtedly in need of integrative/critical reviews. In this regard, the authors did an excellent choice of topic, and they also did a massive work of literature review. I think, however, that the main argument (prosodic processing is missing in the investigation of neural entrainment to speech) is not consistently presented, partly due to lack of clarity in concept definition. I also found that the paper seems to lack direction – as if it was built from a concatenation of fragments, instead of following a clearly defined plan, both at the micro and macrolevels. Overall, I am afraid this review may confuse the reader more than inform him/her. Since I am not sure that these structural problems can be easily revised (specially within the time frames of this journal), I would suggest rejection of this paper. I hope this does not discourage the authors from moving on with this potentially valuable work.
Lack of clarity in concept definition
Speech amplitude envelope – It is defined as “the broadband contour of fast-moving spectral content (“fine-structure”), and it encapsulates important temporal features of the acoustic signal” (ln 68). Ahead, they mention that it captures “features of timing, pitch and loudness” (ln 80). My opinion is that this definition is vague, since it does not clearly exclude anything in the signal. The problem starts with “fast-moving spectral content”: what does it mean – F0 and higher-than-F0- frequencies, including formants and formant transitions (spectrum in a strict sense)? Does it relate to segmental information (implying formant and formant transition detection), suprasegmental (relying on F0), both?
Spectral vs. temporal information – Related to the previous, definitions for these terms are never provided, although they are a target for discussion (ln 76-78). Moreover, how do spectral and temporal levels relate to the speech envelope? Does the latter contain both (see ln 73-81)?
Prosody – Although the authors provide a definition early in lns 34, assigning three dimensions to prosody (stress, intonation and rhythm), in the remainder of the text they refer to prosody as an apparently unidimensional object. This raises obstacles in the understanding of the main argument, as I will try to show next.
Main argument not consistently presented
-The main argument is that prosodic processing is absent in the investigation of neural entrainment to speech. However, the authors mention that the speech amplitude envelope reflects stress, syllable rate, and phonemic structure (lns 82-87). Getting back to the authors’ definition of prosody, I would say that prosody is inherently targeted by studies of entrainment to speech envelope since stress and rhythm (syllable rate) are conveyed by the latter. This is stated explicitly in lns 90-91 (see also ln 241, where the authors write that the speech envelope reflects rhythm). The exception would be intonation. Could it be that the authors are referring to the intonation dimension of prosody, when they refer to the “missing link” (suggested perhaps in lns 100-101)? If it is so, I guess it should a priority to make it clear. If it is not the case, then the argument does not make much sense, unless I am seriously missing something (this is a possibility that I leave open to the editor’s judgement).
-In ln 58, the authors state their argument and say that the speech envelope may be linked to prosody contour. Again, what do they mean by prosody contour? Is it intonation? Because if the term relates to all prosodic aspects, we will soon be presented with findings that the speech envelope relates to stress and rhythm (lns 85-87), providing the answer to the authors’ question.
-the idea of entrainment to different frequency bands, reflecting different linguistic levels, is a core aspect in this review. However, there are apparent inconsistencies which are never discussed (or else non-clarified). The authors start by defining a band in the speech envelope below 4 Hz to establish the realm of prosodic traces (lns 90-91), and another from 8 to 50 Hz to define phonemic content (ln 87). Later on (ln 111), they say that acoustic-phonetic processing occurs in lower frequencies (<8Hz). How does this all integrate? Are the authors referring to something other than neural entrainment to the speech envelope in lns 111?
Lack of direction (micro and macro)
-I guess the obvious structure for a paper like this would be (a) to show that studies of neural entrainment to speech envelope have contributed to understand a number of processes in speech comprehension but (b) they have failed to do so concerning prosodic processes (or is it intonation?...), even though it should be possible. The authors move away from this obvious structure in several ways, making it very difficult to follow the big picture. The main argument (see above) is presented several times, under different shapes and contexts (e.g., ln 58, ln 100, lns 244-251, 338-339,340-344), and it is hard to understand what motivates the question subtending the main argument. At the end of the day, it feels like asking: what have researchers done till now? What have they missed? Did they actually miss prosody? Is the problem that the authors identify out actually a problem?
-instead of using the obvious structure (see above), the authors expand into topics such as the effects of attention or cross-modal integration in neural entrainment: how does this relate to the missing link of prosody? They also emphasize the importance of neural entrainment in a number of pathologies, but the extent to which these relate to prosody is unclear.
-at the microlevel, I found a number of gaps in the authors text sequencing and also instances of poor phrasing, such as:
Ln 35: prosodic cues contribute affect? – typo?
Ln 45: is it due to the suprasegmental nature of prosody that one extracts a wealth of information? – poor phrasing
Ln 48: the topic of child-directed speech is introduced abruptly. Maybe the authors want to present it as an illustration of the role of prosody, but it does not come like that to the reader (sequencing).
Ln 55- these acoustic properties….Which properties? I guess they are in ln 47, but they are too far away at this point (sequencing).
Ln 58 -60: The authors argue something related to the speech envelope without having mentioned this term before (sequencing).
Ln 73-75: I cannot understand this sentence, namely the word “but” (sequencing).
I hope my comments will help the authors to publish in the near future, and I wish them the best of luck.
Author Response
This was an extremely well-written review and I commend the authors for their extensive analysis, appropriate and thorough citations, and choice in figures to illustrate their points. Their assessment of the current trends in the literature and suggestions for future directions were both impressive and inspiring to me as a researcher in this field. I have very few suggestions as this was an excellent review paper.
Thank you for these kind comments.
1) While not the primary purpose of the paper, I personally would have like to see a bit more discussion regarding the importance of the speech envelope as it relates to the processing of emotional stimuli, particularly in relation to the developmental and clinical relevance section of the review. Two or three sentences would suffice.
We have added a few sentences discussing emotional prosody and the need for future research to address this (ln 342-346).
2) While the delta band/frequency and its relationship to cognitive processing are often referred to in the literature, a sentence describing it in lay terms may be helpful for readers that are less well-versed in the subject.
We have added a few sentences describing the delta band in hopes of clarifying this issue (ln 277, 305).
2) The sentence on the last line of page 2 that begins with "Naturally" and continues to lines 74 and 75 on page 3 is a bit unclear. I believe the authors would better make their point if they clarified that detecting frequency modulations is possible from AM alone (e.g. add the word alone or solely).
We have reworded this statement to clarify our meaning (ln 87).
3) This may be my eyes playing tricks on me, but it appears that lines 187-191, beginning with the sentence that starts "Ding and Simon", is a different or smaller font than the rest of the paper.
Thank you for noticing this. That section was in a different font, and we have corrected it.
Again, this was a very well-written paper and I applaud the authors for their efforts in making a complicated subject clearer for the rest of us.
Reviewer 3 Report
This was an extremely well-written review and I commend the authors for their extensive analysis, appropriate and thorough citations, and choice in figures to illustrate their points. Their assessment of the current trends in the literature and suggestions for future directions were both impressive and inspiring to me as a researcher in this field. I have very few suggestions as this was an excellent review paper.
1) While not the primary purpose of the paper, I personally would have like to see a bit more discussion regarding the importance of the speech envelope as it relates to the processing of emotional stimuli, particularly in relation to the developmental and clinical relevance section of the review. Two or three sentences would suffice.
2) While the delta band/frequency and its relationship to cognitive processing are often referred to in the literature, a sentence describing it in lay terms may be helpful for readers that are less well-versed in the subject.
2) The sentence on the last line of page 2 that begins with "Naturally" and continues to lines 74 and 75 on page 3 is a bit unclear. I believe the authors would better make their point if they clarified that detecting frequency modulations is possible from AM alone (e.g. add the word alone or solely).
3) This may be my eyes playing tricks on me, but it appears that lines 187-191, beginning with the sentence that starts "Ding and Simon", is a different or smaller font than the rest of the paper.
Again, this was a very well-written paper and I applaud the authors for their efforts in making a complicated subject clearer for the rest of us.
Author Response
This review aims to highlight the role of prosody in speech perception and the lack of empirical neurophysiological investigations into its neural underpinnings. In particular, the review focuses on the use of envelope tracking measures as a means to probe natural speech processing in electrophysiology. The paper reviews the recent envelope tracking literature and the resulting advances in our understanding of speech processing as well as selective attention and multisensory integration, linking it to prosodic information processing. Finally, the paper discusses the potential of this approach to elucidate on the neural underpinnings of various developmental and clinical disorders and discusses the future directions that the field should take in order to dos so, bearing in mind the role of prosody.
The role of prosody perception in speech processing has indeed been much neglected in many studies that examining neural entrainment to the speech envelope. But its importance to speech perception is in no doubt, thus this review represents an important call to researchers to fill in the gaps in our knowledge regarding how prosody contributes to envelope entrainment. The review starts off very strong and is extremely well written, emphasizing the relevance of the article quite well. The first section on prosody perception sets up the review nicely, is informative and has a really nice flow. I really enjoyed reading it and learned quite a bit! However, I am regretful to say that the quality of the review declined from there on for reasons that I outline below. As such, I recommend that the manuscript is rejected in its current state but hopefully can be accepted at a later date with some major revisions. I have given as much advice as I can in the minor comments section in the hope that it will help the authors improve the manuscript.
Thank you for this feedback. We have significantly revised the manuscript based on the following comments, and we feel that we now have a much better structure that strongly supports our main argument.
Major comments
As indicated by the number of minor comments below, the section on Neural Entrainment to the Speech Envelope in particular, contained many inaccuracies in terms of the literature and terminology which did not match the quality of the previous sections. I would suggest some careful reading of seminal reviews on encoding/decoding models and system identification by the likes of (Wu et al., 2006; Haufe et al., 2014; Holdgraf et al., 2017).
Thank you for this comment and for the related minor comments below. We have made edits as recommended when appropriate. We elected to include a brief introduction to the wide array of methodologies used because an otherwise in-depth discussion is beyond the scope of this paper and is available elsewhere (Haufe et al., 2014; Crosse et al., 2016; Holdgraf et al., 2017; Ding & Simon, 2012).
I felt there were large sections of the article that got lost in terms of direction. As if they were written as part of something else and joined together with a token unifying statement at the end to bring the reader’s attention back around to prosody perception. I feel the main argument of the article needs to be better woven into the fabric of all the sections, and not just the first. This may take substantial rewriting but will be worth it and improve the quality and flow of the article immensely. Look no farther than the first section for inspiration.
While the main aim of the review was to call researchers to call researchers to consider the role of prosody in their investigations, I felt that the authors could have done more to provide details as to how this could be achieved, particularly in the final section. Proposing specific details about future approaches would greatly enhance the value of such work. What kind of empirical evidence is needed? Are there any theoretical or computational models that would be useful to consider?
We have revised the paper throughout to have a coherent flow that consistently supports the purpose of the paper, which is—as you pointed out—to encourage researchers to consider the role of prosody in studies of speech envelope entrainment. While there are potential methodological strategies that could be used to target prosody, we feel that the theoretical consideration is the primary piece that is lacking in the current literature, and we hope that we have made that clear in our revised draft.
Minor comments
There is no reference to the table in the text. Also, paper by Crosse and Lalor (2014) retracted as effect was due to systematic error.
We have added a textual reference to the table (ln 262), and we have removed Crosse and Lalor (2014) from the table.
Abstract: try to link the second half of the abstract which talks about envelope tracking back to prosodic processing.
We have revised the abstract as suggested.
In the speech literature, the amplitude envelope is more commonly referred to as the temporal envelope, in order to distinguish it from a spectrogram representation. The fact that it reflects fluctuations in intensity or amplitude is a given.
The envelope is studied in many disciplines and is associated with various nomenclatures. We have acknowledged that temporal envelope is also a common term for it (ln 72).
Line 82: What do the authors mean by a linear representation? Are they referring to the fact that the broadband envelope is the sum of the envelope at each frequency band?
Yes, envelopes are commonly referred to as linear mappings of parameters, such as numerous frequency bands.
Line 87-88: What do the authors mean by energy in the power spectrum below 4Hz was intercorrelated? As in the power at different narrowbands below 4Hz?
Yes, and we have clarified this in the text (ln 102).
Line 111-112: This statement contradicts your earlier statement where you relate the temporal scale of linguistic units to particular neural oscillatory frequency bands. My understanding is that higher frequencies reflect processing of more fine-grained segmental units (phonemes, morphemes, syllables etc.) and lower frequencies more course-grained suprasegmental units (words, phrases, sentences etc.). This also makes sense when you think about how ERP research has characterized the timecourse of speech evoked AEPs, relating longer latencies (which correspond to lower frequencies) to higher-level linguistic processing (phonetic, lexical-semantic etc.). See Salmelin (2007).
Yes, we agree with this explanation, and we have described it as such in the manuscript (ln 102-105).
Line 119: I’m not sure if I would use the term ‘periodic qualities’ to describe the properties of speech that neuronal populations entrain to. While the segmental and suprasegmental units of speech tend to have specific temporal properties, they do not unfold within a given sentence in a periodic manner necessarily, even if they always maintain their temporal properties. The rhythm of speech is highly variable, but at the same time predictable due to context and prior knowledge. I would use the term ‘temporal properties’ instead.
We do not claim that speech is periodic (we agree that the rhythm of speech is highly variable). Here we are referring to the actual periodic oscillations that make up sound. The definition of phase locking depends on the presence of these periodic qualities, so we feel that this is the accurate term to use in this case.
Line 119-121: What about an evoked ac
This sentence does describe an evoked response.
Line 137-139: Also, non-invasive (EEG) studies have shown that use of more complex non-linear approaches does not improve modal performance significantly (Crosse, 2011; Power et al., 2011a, b) which may be due to the diversity of non-linear responses across neuronal populations making them less detectable by non-invasive recording such as EEG (Mesgarani et al., 2009; Crosse et al., 2016b).
Thank you for mentioning this, but we feel this may be a branch away from the scope of this paper.
Line 143-144: Multivariate reconstruction filter has to be estimated first using backward modeling approach before applying to new EEG data to reconstruct the envelope.
Thank you for mentioning this, but we feel this level of information is outside the scope of this paper. It is our intention that any readers who would like more information on this modeling approach will refer to the papers that we cite.
Line 143-145: Reconstruction accuracy can be measured a number of different ways, not just Pearson’s r. Could use different measures of correlation (e.g., Spearman, Kendall, Rankit) and also error measures such as root mean squared error (RMSE). Better to say “Reconstruction accuracy can be measured by Pearson’s…”.
We have rephrased this statement as suggested (ln 172 and 181).
Line 146-148: This sentence says the same thing twice using different language. Forward modeling is the approach, and TRF is what quantitatively describes the mapping. TRF can also be referred to as a forward model. Note difference between ‘a forward model’ and ‘forward modeling’.
Thank you for mentioning this, and we feel that this level of redundancy is helpful to lay readers who are unfamiliar with this methodology.
Line 155: Backward models do not use ‘TRFs’. It’s the same mathematics, but with the input and output reversed. Because TRF stands for temporal response function, by definition, it refers to the impulse response of the system, which can only be described in the forward direction as that is how the response is generated. Backward models (or reconstruction filters or decoders), are just mathematical representations of how the stimulus can be inferred from the responses, but in reality the responses do not cause or generate the stimulus. It’s just a useful method for quantifying stimulus encoding.
We have rephrased this sentence as suggested (ln 174).
Line 1156-157: I wouldn’t use the term generate when describing backward modeling to avoid confusion with forward models which are by definition generative models. Maybe use ‘yields’ or ‘derives’.
We have rephrased this sentence as suggested (ln 175).
Line 158: Correction: “this decoder is derived by minimizing the mean squared error between the stimulus and reconstruction”.
We have rephrased this sentence as suggested (ln 177).
Line 160: It’s not the contribution-dependent channel weighting that “does not allow inter-channel redundancies”, it’s the fact that in the backward model, the autocovariance structure is now based on the response (RTR), meaning the inter-cannel correlations are divided out of the model, revealing only unique contributions at each channel.
We have rephrased this sentence as suggested (ln 179).
Line 162-163: Again, there are a number of ways to evaluate reconstruction accuracy, not just Pearson’s r.
Thank you for mentioning this again. We feel that the current wording is not exclusive of other possibilities. Our intention is to give a brief overview of this method rather than an in-depth tutorial.
Line 163: I would use “reliable index” as opposed to “efficient representation”.
We have rephrased this sentence as suggested (ln 182).
Line 148-169: I agree, see Ding et al. (2014) theory of an analysis-by-synthesis mechanism and its relationship to envelope tracking.
Thank you for mentioning this, and we do include this study later in the paper, as well as in Table 1.
Line 178-179: True, but there are studies demonstrating strong relationships between envelope tracking and speech intelligibility (Ding and Simon, 2013). So while entrainment may be primarily driven by acoustic features, it is also enhanced by intelligibility in both a bottom-up manner (signal quality, background noise) and a top-down manner (prior knowledge, attention).
Yes, we agree that entrainment has bottom-up and top-down aspects, and we present both in this paper. We have rephrased lines 200-205 to reiterate this.
Line 186: “Neural circuit” does not convey the broad, distributed network of brain areas that are likely involved in speech processing and auditory scene analysis.
We agree, but we do not intend to go into that amount of detail in this sentence.
Line 196-199: These inverted peaks that suppress the unattended speaker have also been demonstrated by O'Sullivan et al. (2015).
We have added this study to the citations in this section (ln 216).
Line 200: I wouldn’t refer to the forward modelling approach in Power et al. (2012) as AESPA, this term was used by the authors because of their previous research using white noise (spread spectrum) stimuli.
It’s the exact same thing as a speech envelope TRF so can be referred to as such.
We have removed the term AESPA as requested.
Line 201-202: O'Sullivan et al. (2015) did not replicate the study by Power et al. (2012), they actually re analyzed the data from the latter study, but using a backward modelling approach instead of forward modelling.
We have removed this section, as it was tangential to the main argument.
Line 225: I wouldn’t describe this as the “summation of the two modalities” but rather just a combined unisensory or non-integrative multisensory account.
We have clarified this point (ln 236).
Line 227-229: Why a distinct mechanism? Congruent lip movements also implicitly encode information about the acoustic envelope (Crosse et al., 2015b; O’Sullivan et al., 2017) that is highly predictive of intelligibility or lip-reading ability (Crosse et al., 2015a). Why would this not be a similar process to auditory envelope tracking just within a different sensory system?
Thank you for mentioning this. We used the word “distinct” to clarify that the visual system has its own separate processing mechanism. However, you are correct that it is a similar process only in a different modality. We have changed the wording to reflect this (ln 239).
Line 230-231: How can A and V signals interacting dynamically be a discrete process?
The discrete process is that of “multisensory integration”, which is its own area of study and is cited here, but is beyond the scope of this paper to cover in detail.
Line 246: Certainly from a cross-modal perspective, the role of prosodic information has been highlighted. Munhall et al. (2004) showed that delta frequency head movements, which have been shown to convey prosodic information, are important to speech intelligibility and Megevand et al. (2018) recently showed delta-band phase resetting of auditory cortical oscillations by visual speech.
Other studies (Ding and Simon, 2013; Ding et al., 2014; Crosse et al., 2016a) have demonstrated the robustness and importance of delta-frequency activity (which likely reflects prosodic information) for entraining to speech-in-noise.
Thank you for mentioning this. Your statement is exactly the motivation for this paper—many studies have focused on delta-band entrainment to speech, which reflects prosodic information, yet prosody is not overtly discussed in these studies. The purpose of our literature review is to show that prosody has been an important piece of the literature, yet it so often goes unacknowledged.
Line 276: See also Foxe et al. (2015) for compelling example of AV speech integration deficit in ASD.
Yes, this is another fine example.
Line 276-278: This may not be exclusively related to the so-called temporal binding hypothesis. It could be do with disrupted connectivity (Vasa et al., 2016), impaired cross-modal inhibition (Murphy et al., 2014) or even a delay in the developmental shift from multisensory competition to multisensory facilitation (Cuppini et al., 2018) due to lack of multisensory experience (Cuppini et al., 2017).
Yes, we agree that there are a number of possibilities to explain the deficits seen in ASD, and for the purposes of this paper, we present a brief overview from the angle of neural entrainment. We have revised the text (ln 292-294).
Line 291-293: It’s not clear to me why considering “cross-frequency neural synchronization” would “help identify the neural underpinnings of impaired phonology” in dyslexia.
We agree that this sentence was poorly placed, and we have revised it (ln 305-307).
Line 316: typo
We have thoroughly reviewed this line and the lines surrounding it, and we are unable to detect any typographical errors. Perhaps the reviewer is referring to the word “thereof”, but this has been removed in our edits.
Line 320-321: See new empirical evidence in support of this using psychophysics (Fiebelkorn et al., 2013) and electrophysiology (Fiebelkorn et al., 2018; Helfrich et al., 2018).
Thank you for mentioning these studies for additional support.
Line 336: “hierarchically-organized time scales”.
We have made this edit (ln 353).
Line 341: Can the authors expand on how to consider this relationship.
We have expanded on this to clarify our point (ln 358).
Line 346: “relationship between”
We have clarified this sentence (ln 366).
Round 2
Reviewer 2 Report
Brainsci-429009 R1
Review
The authors have successfully solved some problems in the manuscript, and I congratulate them for that.
However, I still found myself struggling with the main argument. Namely, in their response to my comments, the authors stated that the current version of the paper aims
(c) to give an overview of studies that use the envelope and demonstrate that
prosody may be a key factor even if not explicitly mentioned, and
(d) to demonstrate that prosody has a clear role in these studies and should be addressed more explicitly in the literature.
Concerning c), I still do not understand
-why prosody has not been explicitly mentioned in envelope studies. For instance, Goswami and colleagues (which the authors cite), have developed a whole research line based on the possibility that dyslexics have entrainment deficits at the stress and/or syllable rate entrainment levels. This is prosody.
-what do they mean by “explicitly mentioned”?
-prosody is not explicitly mentioned as opposed to what explicitly mentioned dimensions? Which dimensions have been explicitly mentioned and addressed, other than prosody?
Concerning d),
-what do you mean by “prosody has a clear role in these studies and should be addressed more explicitly”? That neural entrainment patterns reflect prosodic patterns but authors have interpreted these entrainment patterns otherwise?
-what do you mean by “address more explicitly”? Provide prosody-only stimuli? Interpret entrainment data in light of prosodic properties?
Although this is just one problematic issue, this is a major one, since it is about the direction and purpose of the paper. That is why I am suggesting major revisions.
In case the authors see no way of being straightforward about this, I would suggest that they leave out the prosodic-missing-link argument and present the paper simply as a review of speech entrainment studies, because that could be enough.
Author Response
The authors have successfully solved some problems in the manuscript,
and I congratulate them for that.
However, I still found myself struggling with the main argument. Namely,
in their response to my comments, the authors stated that the current
version of the paper aims
(c) to give an overview of studies that use the envelope and demonstrate
that prosody may be a key factor even if not explicitly mentioned, and
(d) to demonstrate that prosody has a clear role in these studies and
should be addressed more explicitly in the literature.
Concerning c), I still do not understand
-why prosody has not been explicitly mentioned in envelope studies. For
instance, Goswami and colleagues (which the authors cite), have
developed a whole research line based on the possibility that dyslexics
have entrainment deficits at the stress and/or syllable rate entrainment
levels. This is prosody.
-what do they mean by “explicitly mentioned”?
-prosody is not explicitly mentioned as opposed to what explicitly
mentioned dimensions? Which dimensions have been explicitly mentioned
and addressed, other than prosody?
Yes, we agree that some envelope studies have explicitly discussed prosody. Our intention was not to say that prosody is never mentioned and we agree that some studies have explicitly focused on or considered prosody. We meant to convey that even when prosody is not a primary focus of a study, it may still contribute to the findings (e.g., including in studies focusing on syntactical features or considering the roles of intelligibility, attention, audiovisual integration, etc.). Our contention is that prosodic features should be considered as part of these research questions and stimuli, as well. We have adjusted the language throughout and elaborated upon this in our revisions (lines 320-325).
Concerning d),
-what do you mean by “prosody has a clear role in these studies and
should be addressed more explicitly”? That neural entrainment patterns
reflect prosodic patterns but authors have interpreted these entrainment
patterns otherwise?
Yes, authors have interpreted entrainment patterns in terms of intelligibility, attention, audiovisual context, and at various levels of linguistic units, and we recognize that these components are important for our understanding of entrainment. We note that prosodic features are ubiquitous in all natural speech stimuli; thus, researchers may need to consider the role of prosodic features in these different experimental conditions. We describe this in our revisions (lines 350-361).
-what do you mean by “address more explicitly”? Provide prosody-only
stimuli? Interpret entrainment data in light of prosodic properties?
We agree that these are potential avenues for future research. We have elaborated on potential ideas for future work in the addition of Table 2 (line 333).
Although this is just one problematic issue, this is a major one, since
it is about the direction and purpose of the paper. That is why I am
suggesting major revisions.
In case the authors see no way of being straightforward about this, I
would suggest that they leave out the prosodic-missing-link argument and
present the paper simply as a review of speech entrainment studies,
because that could be enough.
Again, we apologize for these confusing statements in our previous response, and thank the reviewer for their thoughtful comments. We hope that our current revisions provide a more clear review of the literature and potential directions and considerations for future research.